# Personalized Healthcare: The Importance of Patients’ Rights in Clinical Practice from the Perspective of Nursing Students in Poland, Spain and Slovakia—A Cross-Sectional Study

**DOI:** 10.3390/jpm11030191

**Published:** 2021-03-11

**Authors:** Ewa Kupcewicz, Elżbieta Grochans, Helena Kadučáková, Marzena Mikla, Aleksandra Bentkowska, Adam Kupcewicz, Anna Andruszkiewicz, Marcin Jóźwik

**Affiliations:** 1Department of Nursing, Collegium Medicum, University of Warmia and Mazury in Olsztyn, 14 C Zolnierska Street, 10-719 Olsztyn, Poland; 2Department of Nursing, Pomeranian Medical University in Szczecin, 48 Zolnierska Street, 71-210 Szczecin, Poland; grochans@pum.edu.pl; 3Department of Nursing, Faculty of Health, Catholic University in Ruzomberok, 48 A. Hlinku Street, 034-01 Ruzomberok, Slovakia; helena.kaducakova@ku.sk; 4Department of Nursing, Campus de Espinardo, University of Murcia, Edificio 23, 30100 Murcia, Spain; marmikla@yahoo.com; 5Murcian Institute of Biosanitary Research (IMIB), 30120 Murcia, Spain; 6Oncological and General Surgery Clinic, University Clinical Hospital in Olsztyn, 30 Warszawska Street, 10-900 Olsztyn, Poland; a.bentkowska.kruszwicka@gmail.com; 7Faculty of Law and Administration, University of Warmia and Mazury in Olsztyn, 2 Oczapowskiego Street, 10-719 Olsztyn, Poland; adam.kupcewicz@gmail.com.pl; 8Department of Basic Clinical Skills and Postgraduate Education for Nurses and Midwifes, Nicolaus Copernicus University in Toruń, Łukasiewicza 1 Street, 85-821 Bydgoszcz, Poland; anna.andruszkiewicz@cm.umk.pl; 9Department of Gynecology and Obstetrics, Faculty of Medicine, Collegium Medicum, University of Warmia and Mazury in Olsztyn, 44 Niepodleglosci Street, 10-045 Olsztyn, Poland; marcin.jozwik@uwm.edu.pl

**Keywords:** patients’ rights, student, nursing, personalized medicine

## Abstract

Background: This study aimed to define the role and importance of patients’ rights in personalized healthcare from the perspective of nursing students in Poland, Spain and Slovakia. Methods: The research was carried out by means of a diagnostic survey, using the survey technique, with the participation of 1002 nursing students attending a full-time undergraduate study program at three European countries. The “Patients’ rights” questionnaire was used as a research tool. The average age of students was 21.6 years (±3.4). The empirical material collected was subjected to a statistical analysis. Results: The study demonstrated that 72.1% of nursing students from Spain, 51.2% from Poland and 38.5% from Slovakia believe that patients’ rights are respected at a good level in their country. Significant intergroup differences (F = 67.43; *p* < 0.0001) were observed in the self-assessment of students’ knowledge of patients’ rights. The highest average values were obtained by students from Spain (3.54 ± 0.92), while 35.9% of students from Slovakia and 25.5% from Poland were quite critical and pointed to their low level of knowledge of patients’ rights in their self-assessment. When ranking patients’ rights related to respecting dignity, students from Spain obtained much higher average values (4.37 ± 0.92) than students from the other two countries. Conclusions: The level of students’ knowledge of patients’ rights and the respect for patients’ rights by medical personnel is, in the opinion of the respondents, quite diverse and requires in-depth educational activities among nursing students at the university level in respective countries.

## 1. Introduction

Personalized medicine is a model in which disease prevention and treatment is based on the patient’s unique clinical, genetic and environmental characteristics [1,2,3]. Personalized healthcare, providing opportunities for a more precise approach to individual medical care, is of particular benefit to the patients. It also poses a unique challenge in terms of its holistic approach to health and sickness. This approach assumes that it is a person who should be treated, not just an illness, since a strong link exists among body, soul and mind; they form one entity and only a balance among them can ensure the state of health [4]. 

Personalized healthcare involves the important issue of patients’ rights, which determine the status of the patient during the provision of health services and the obligations of the medical personnel towards the patients as well as towards their relatives [5]. Consequently, the observance of patients’ rights by medical personnel in clinical practice is regarded as an ethical obligation and a legal obligation [6]. The concept of patients’ rights was developed on the basis of the Universal Declaration of Human Rights adopted in 1948 by the United Nations (UN) General Assembly, which explicitly states that every human being has an inherent right to life, freedom, privacy, free development in society and respect for their dignity [6,7,8]. The aim of the concept of patients’ rights is to protect the autonomy of the patient from interference by others, as well as the right to demand the rightful conditions for the exercise of those rights [6]. 

According to the World Health Organization (WHO), patients’ rights vary from country to country, and it is often the prevailing cultural and social norms that determine the catalogue of patients’ rights applicable in a given country [7]. However, there is international consensus that all patients have a fundamental right to privacy, to the confidentiality of their medical data, to consent or refuse treatment and to information about the risks associated with medical procedures [7].

In Europe, the observance of patients’ rights is guaranteed, among others, by the Convention on Human Rights and Biomedicine of 1997 (also referred to as the European Bioethics Convention or the Oviedo Convention) [9]. Another document is the European Charter of Patients’ Rights, issued in 2002 by the Active Citizenship Network, which governs basic issues concerning patients’ rights [10]. The charter mentions, among others, the right of access to health services and the right to respect of patients’ time, regardless of the phase or place of treatment, which states that every person has the right to receive the necessary treatment within a swift, predetermined period [10]. Such a guarantee has been introduced in selected European countries, e.g., Sweden, Denmark, Finland, Norway, England, Scotland, Wales, Ireland, Portugal, Spain and the Netherlands [11]. The European Parliament and the Council of the European Union have played a significant role in protecting patients’ rights, recognizing that the Member States of the European Union have a responsibility to provide citizens in their territory with safe, efficient, high-quality and quantitatively adequate medical care. The undertaken actions resulted in the introduction of Directive 2011/24/EU of the European Parliament and of the Council of 9 March 2011 on the application of patients’ rights in cross-border healthcare [12,13,14,15,16]. As numerous studies have shown, the management of patient care with regard to personal needs, rights and duties requires a certain degree of personalization [17,18]. It is connected with the theoretical and clinical preparation of students of nursing studies for future work related to patient care [19]. Nursing program curricula provide students with the opportunity to achieve learning outcomes in terms of knowledge of human rights, children’s rights and patients’ rights [19,20]. In the process of socialization under the guidance of academic teachers, nursing students, as future nurses, acquire social competences to be guided in their future work by professional values when making decisions in the face of emerging, healthcare-related ethical challenges [21]. In terms of social competence, a graduate of nursing studies is ready to respect the rights of the patient, to respect the dignity and autonomy of the persons under their care, to be guided by the welfare of the patient and to show understanding for differences in worldview and culture and empathy in relation to the patient and their family [19,20]. 

The aim of this study was to define the role and importance of patients’ rights in personalized healthcare from the perspective of nursing students in Poland, Spain and Slovakia.

The following research problems were formulated:Are there differences in the observance of patients’ rights in healthcare-providing institutions in the opinion of nursing students in Poland, Spain and Slovakia and to what extent?To what extent does knowledge of selected patients’ rights in clinical practice regarding an ill or healthy person differ among nursing students in Poland, Spain and Slovakia?

## 2. Materials and Methods 

### 2.1. Settings and Design

The study was carried out between May 2018 and April 2019 by means of a diagnostic survey, using the survey technique, with the participation of 1002 nursing students, studying in first degree (bachelor’s degree) programs in a full-time system at the University of Warmia and Mazury in Olsztyn and the Pomeranian Medical University in Szczecin (Poland), the University of Murcia in Murcia (Spain) and the Catholic University in Ružomberok (Slovakia). The surveys were carried out at the place where the didactic classes for students were conducted, and the distribution of the prepared sets of paper questionnaires to a given university was handled by one of the researchers. Upon obtaining permission from the academic teacher to conduct the survey, students were informed of the purpose and the scope of the study and provided with instructions on how to complete the questionnaire. Students had the opportunity to ask questions and receive comprehensive explanations. The survey was anonymous and voluntary; the time taken to complete the questionnaires in person was approximately 20 min. The inclusion criterion for the students in the study was the age of the subjects up to 30 years, while the exclusion criterion was the absence of informed consent to participate in the study. Students could also opt-out of the study at any time without providing a reason. In total, 1017 survey forms were distributed among students. After collecting data and eliminating defective questionnaires, 1002 (i.e., 98.5%) correctly completed paper version questionnaires were accepted for the final statistical analysis. The collected data were entered into a spreadsheet in Excel software and the results were analyzed collectively. 

### 2.2. Participants

The investigated group included 404 (40.3%) students from Poland, 208 students (20.8%) from Spain and 390 students (38.9%) from Slovakia. The mean age for all subjects was 21.6 years (±3.4). Among the students, women accounted for 91.3% (*n* = 915), men for 8.7% (*n* = 87). The distribution of first-, second- and third-year students across the universities was similar. The most numerous group were second-year students (*n* = 458; 45.71%), while 329 (32.83%) studied in the first year and 215 (21.46%) in the third year. The age of the students was analyzed in three age groups, assuming the following ranges: ≤20 years (*n* = 401; 40.02%), 21–22 years (*n* = 410; 40.92%) and ≥23 years (*n* = 191; 19.06%). The presented data are part of a larger international project and detailed sociodemographic characteristics are also included in other publications [22,23]. 

### 2.3. Research Instruments

A structured survey questionnaire created by the authors, entitled “Patients’ Rights”, was used to measure the variables of the study. The questionnaire consisted of two parts. The first part contained subjective questions to determine the structure of the surveyed group of students in terms of sociodemographic variables such as place of residence (country), gender, age, level of education and mode and year of studies. These questions included five closed questions, including all possible answer options, and one open question to determine the age of the respondents. The second part of the questionnaire contained 14 questions of an objective or subjective nature, which made it possible to determine the level of students’ knowledge of patients’ rights and selected aspects related to their observance in personalized healthcare addressed to sick and healthy people. The questions included two so-called “ranking” questions, to self-assess students’ level of knowledge of patients’ rights and to prioritize the patient’s right to dignity. In the first ranked question, the respondent could choose from a rating scale from 2 to 5, reflecting the level of their current knowledge of patients’ rights, where “2” and “3” indicated a low level of knowledge, “4” an average level and “5” a high level. Similarly, in the second question concerning the ranking of the importance of the patient’s right to respect for dignity, the respondent indicated on a rating scale from 2 to 5, the rank given to the patients’ right, where “2” and “3” indicated a low rank, “4” an average rank and “5” a high rank. In the remaining questions, the respondent was asked to mark one of four or five possible answers for each question.

The process of constructing the applied tool involved the development of a set of statements concerning the studied variables using information retrieved from the literature on the subject. Once the final set of questions in the Polish language was established, it was translated into the Spanish and Slovak languages. The research tool in equivalent language versions was subjected to a psychometric assessment. The reliability of the questionnaire was assessed through the internal consistency estimation, which was established based on Cronbach’s alpha coefficients. When estimating the internal consistency degree, two of the questions from the second part of the questionnaire were rejected due to only slight thematic coherence. The reliability of all the other questions, measured by the value of the Cronbach’s alpha coefficient, ranged from 0.60 to 0.71 [24].

### 2.4. Statistical Analysis

Statistical analyses were performed using the Polish version of STATISTICA 13 (TIBCO, Palo Alto, CA, USA). The mean, standard deviation and confidence interval for the mean ±95%, median, minimum and maximum were used to describe some of the analyzed variables. The ANOVA analysis of variance (F-test) comparing multiple samples of independent groups was used to investigate the significance of differences in the ranking of the subjective assessment of students’ level of knowledge and in the ranking of the patients’ right to respect for dignity. The significance of variation in the knowledge of patients’ rights was assessed with the chi-square test (χ^2^). For all tests, a significance level of *p* < 0.05 was assumed. The analyses of the results are presented in descriptive, tabular and graphical forms [24]. The research meets the criteria for a cross-sectional study [25].

## 3. Results

Students participating in the survey were given the opportunity to express their opinion on selected patients’ rights in personalized healthcare based on their own experience of staying in a healthcare facility as a patient and as a nursing student, deepening their knowledge during clinical activities. The majority of nursing students (79.8%; *n* = 800) had used medical services as a patient in a hospital or clinic/outpatient clinic in the last three years preceding the survey. Their level of satisfaction with the quality of the medical services provided at that time significantly varied (χ^2^ = 45.53; *p <* 0.0001). The majority of students expressed a positive opinion on the overall quality of the medical services provided. However, only 33.9% of respondents reported that they had been informed about their rights and that information about patients’ rights and the Patient Ombudsman was posted in a publicly accessible place. Taking into account the cultural background and the organization of the healthcare system in the different countries, a significant variation in results was observed (χ^2^ = 124.26; *p <* 0.0001) as regards the provision of information to patients concerning their rights. As analyses show, significantly more respondents in Poland (62.4%) than in Slovakia (37.7%) and Spain (16.5%) were informed about their rights when using medical services. When assessing the observance of patients’ rights in personalized medical care, the results of the answers to the question concerning the provision of a sufficient level of intimacy to the patient during the provision of medical services were also sought. As indicated by the data, 60.5% of respondents confirmed that they were ensured good conditions when receiving medical services that minimized the feeling of embarrassment and reduced privacy.

### 3.1. Observance of Patients’ Rights in Personalized Healthcare as Perceived by Nursing Students

In the opinion of nursing students, respect for the patients’ right to receive comprehensive information about their own health condition and planned medical treatment in personalized healthcare significantly varied (χ^2^ = 315.61; *p <* 0.0001) in the analyzed subgroups. A high percentage (92.8%) of nursing students from Spain indicated that the right in question is respected in their country. However, around half of the respondents (51%) in the Slovak group and only 28.5% in the Polish group were of the same opinion. Further analysis involving the issue of compliance with the patients’ right to receive pastoral care during the hospital stay showed statistically significant differences (χ^2^ = 122.24; *p <* 0.0001). More than half of the respondents confirmed this possibility and one in three Spanish students had no opinion about it, while in the Polish group 16.6% (*n* = 67) and in the Slovak group 13.6% (*n* = 68) of nursing students declared that they had no knowledge about it (Table 1).

In the following analyses, an attempt was made to find a subjective assessment by the nursing students concerning the extent to which patients’ rights were respected in healthcare facilities in Poland, Spain and Slovakia. The distribution of the results was significantly different (χ^2^ = 75.26; *p <* 0.0001). It was found that 72.1% of nursing students from Spain, 51.2% from Poland and 38.5% of students from Slovakia rated the respect of patients’ rights as good (Figure 1). On the other hand, one in five students from the Slovak group (20.8%) also indicated a disadvantageous situation for the patient, indicating that the level of respect for patients’ rights was rather low or definitely low (Figure 1).

### 3.2. Differences in Nursing Students’ Knowledge of Selected Patients’ Rights in Personalized Healthcare

Further analyses involved investigating students’ knowledge of a patients’ rights in personalized healthcare to deposit valuable items to a hospital depository during inpatient treatment in a healthcare facility. The analysis demonstrated statistically significant differences (χ^2^ = 121.64; *p <* 0.0001). It was found that the vast majority of Slovak students (91.3%; *n* = 356) positively responded to the question of whether a patient in healthcare facilities has the right to use a depository for the period of hospitalization. One of the most important patients’ rights in personalized healthcare concerns the confidentiality of patient-related information. The data obtained show that 86.5% (*n* = 867) of nursing students confirm that the patient has the right to data protection and confidentiality concerning the information on the patient by the medical staff. This means that all information about the patient’s health condition, the diagnostic, therapeutic, rehabilitation and nursing activities carried out and any other information obtained in connection with the exercise of the medical profession must not be disclosed to any unauthorized persons and should be treated as confidential. The data in Table 1 allow us to conclude that there is a statistically significant variation in the results among students across countries (χ^2^ = 66.13; *p <* 0.0001) in terms of knowledge of the patients’ right to the confidentiality of information. A higher percentage of students from Slovakia (91.3%; *n* = 356) than students from Spain (79.3%; *n* = 165) indicated their knowledge of this right. In certain situations, medical practitioners are obliged to disclose information covered by professional secrecy. It was found that more than half of the nursing students (54.5%; *n* = 546) felt that medical staff could be exempted from professional confidentiality if the information covered could contribute to a risk to the health and life of others. The analysis of opinions concerning the issuance of copies of inpatient/ambulatory treatment records by the medical facility to the patient demonstrated statistically significant differences (χ^2^ = 51.14; *p <* 0.0001). It was found (Table 1) that 75% of the students (*n* = 156) from Spain confirmed that a medical facility is obliged to provide the patient with the records of inpatient or outpatient treatment, while a slightly lower percentage of students from Poland (71.5%; *n* = 289) and Slovakia (67.7%; *n* = 264) were of the same opinion. The situation was slightly different as regards knowledge declared by the students of the patient’s right to be discharged from a hospital at their own request. The data presented show significant variation in responses among students (χ^2^ = 86.00; *p* < 0.0001). The vast majority of respondents from Poland (79.5%; *n* = 321) and Slovakia (75.9%; *n* = 296) stated that a patient has the right to be discharged from hospital on their own request. In contrast, a significant proportion (35.1%; *n* = 73) of Spanish students stated that a patient can only be discharged upon their own request from a hospital if their life is not in danger. Patients’ rights also include the right to withdraw their objection to the donation of organs and tissues. Data analysis (Table 1) showed significant differences among nursing students (χ^2^ = 15.17; *p <* 0.004); the vast majority of students (72.1%; *n* = 150) from Spain confirmed the patient’s right to withdraw their objection to organ and tissue donation, while 29.5% (*n* = 115) of students from Slovakia, 27.0% (*n* = 109) from Poland and 17.3% (*n* = 36) from Spain stated that they had no knowledge of this issue.

In the course of the study, students were asked to make a subjective assessment of their level of knowledge in the field of patients’ rights, using a rating scale from 2 to 5. In a statistical analysis, significant differences in the level of knowledge (F = 67.43; *p <* 0.0001) were observed between students from Poland, Spain and Slovakia. The highest mean values were obtained by students from Spain (3.54 ± 0.92), while significantly lower scores were found for students from Poland (3.00 ± 0.73) and Slovakia (Table 2).

After establishing low, average and high scores, special attention was paid to the proportion of the surveyed students who rated their competence in the area of patients’ rights as low. As it turned out, as many as 35.9% of students from Slovakia, 26.5% from Poland and 14.9% from Spain were quite critical and indicated in the self-assessment a low level of knowledge of patients’ rights (Figure 2).

Subsequent analyses reported statistically significant differences (F = 3.44; *p <* 0.03) in self-reported knowledge of patients’ rights among the age groups of Slovak students. It was proven that students aged 23 years and older received significantly higher mean values in the assessment (3.0 ± 0.68) than students in the ≤20 years age group (2.7 ± 0.67). This is probably linked to the implementation of educational content on the topic of patients’ rights in classes in subsequent years. However, no statistically significant differences in self-assessed knowledge were found among Polish and Spanish students (F = 0.11; *p <* 0.89 vs. F = 0.18; *p <* 0.83) in the respective age groups. Analyses demonstrated that, in the Spanish group, the year of study significantly determined the level of students’ knowledge of patients’ rights (F = 14.68; *p <* 0.0001). Second-year students received higher mean values in the self-assessment (3.8 ± 0.85) than first-year students. Taking into account the year of study in the analyses, no significant differences in self-assessed knowledge of patients’ rights were found among students from Poland (F = 0.16; *p <* 0.85) and Slovakia (F = 0.81; *p <* 0.44).

### 3.3. Assessing the Importance of the Patient’s Right to Dignity in Personalized Healthcare

In further analyses, an attempt was made to rank, in the opinion of nursing students, the importance of the patient’s right to respect their dignity, since the care of people in health and illness should always be based on respect for their dignity, subjectivity and ensuring intimacy when health services are provided by medical personnel. The right to respect for dignity also includes the right to die in peace. When asked what rank on a 2–5 rating scale the students would give to the patients’ right to dignity, significantly different results (F = 133.56; *p <* 0.0001) were obtained, depending on the surveyed students’ country of origin (Table 3).

The analyses showed that Spanish students obtained significantly higher mean values (4.37 ± 0.92) in the ranking of the patient’s right to dignity than students from the other two countries. Subsequent analyses explored the influence of selected sociodemographic characteristics such as age and year of study within the country on the ratings indicated by students regarding the patient’s right to dignity. The analysis found no statistically significant differences in the rank given to the patient’s right to dignity across age groups in any of the analyzed countries. However, in Spain, the year of study was found to significantly (F = 3.72; *p <* 0.03) influence the level of students’ ranking of the patient’s right to dignity. Spanish first-year students gave a significantly higher ranking to the patient’s right to dignity (4.6 ± 0.73) than third-year students (4.1 ± 1.08; *p <* 0.02). In contrast, there were no statistically significant differences in the ranking given to the patient’s right in students from different years of study in Poland and Slovakia. Therefore, it may be concluded that the results of the studies conducted in Poland, Spain and Slovakia confirm the differences in the knowledge of patients’ rights among nursing students, but they require further scientific consideration.

## 4. Discussion

We attempted to determine the role and importance of patients’ rights in personalized healthcare from the perspective of nursing students in Poland, Spain and Slovakia. It was recognized that the individualized nature of medical services involves respecting the rights of the patient, which protect the patient’s autonomy (freedom) from interference by other parties and provides the basis for claiming the legitimate conditions for the exercise of those rights. According to the procedures applicable in a given healthcare facility, the patient should be informed of their rights, which should be recorded in an understandable and legible manner and available in the patients’ areas.

The results of the authors’ own research show that only one-third of the respondents were informed about their rights before admission to hospital or during the provision of health services and that the information on patients’ rights and the Patients’ Ombudsman was posted in a publicly accessible place. The analyses of research results obtained by many other authors quite often reveal an unfavorable situation of the patient concerning their rights in various medical entities operating in the medical services market. For example, a study conducted by Ansari et al. among 500 Iranian patients in inpatient and outpatient care found that 93.5% of them did not receive any information on patients’ rights [26]. Moreover, a study conducted by Egyptian researchers on a group of 514 patients hospitalized at the Minia University Hospital found that about 76% of patients did not know about the existence of the patients’ rights charter and 98.1% of those surveyed said that the medical team did not inform them of the treatment options available [27]. This means that healthcare providers should place greater emphasis on raising patients’ awareness of their rights and involving them in decisions about their treatment choices. Abedi et al. also indicated the need to increase patients’ awareness of their rights during the delivery of healthcare services [28]. As stated by Agrawal et al., to take effective educational measures to improve general awareness not only among patients but also among various stakeholders in the healthcare system, it is important to assess the awareness of hospitalized patients of their rights [29]. As shown by the results of a study conducted by Tabassum et al. in two hospitals in Lahore (Pakistan) from the public and the private sectors, most patients (64%) were not aware of their rights. However, the level of awareness of patients’ rights was higher in patients receiving medical care in a private hospital than in a public hospital [30]. It is also worth referring to the findings of Mohammadi et al. who, in their study, indicated the need to inform patients about ethical and legal issues related to privacy and confidentiality, before and during admission to hospital [31].

As shown by an analysis of the authors’ own study results, significantly more patients in Poland (62.4%) than in Slovakia (37.7%) or Spain (16.5%) were informed about their rights. Other researchers have also attempted to assess existing barriers to compliance with patients’ rights on the basis of a meta-analysis. The most important factors cited as obstacles to respecting patients’ rights included, among others: excessive workload of nurses, staff shortage, organizational factors and a lack of awareness of the patients’ rights charter among patients, nurses, doctors and students [32]. In the current study, the degree of observing patients’ rights in healthcare institutions in the opinion of nursing students in Poland, Spain and Slovakia is significantly different. Almost 75% of Spanish students rated the level of observing patients’ rights in their country as good. The respect for patients’ rights in Polish and Slovak healthcare institutions was rated much lower by the students. Undoubtedly, there is a need to search for subjective and objective factors affecting the level of respecting patients’ rights in medical facilities.

Interesting results were also presented by Mousavi et al. who showed that the rights of patients admitted to the Intensive Care Unit are more affected than those of patients admitted to other hospital wards. They pointed to inadequate nurse-to-patient ratios, socio-economic problems, working hours and high workload in a limited time as the main factors affecting the quality of nursing practice in terms of respecting patients’ rights, among others [33]. In contrast, Waddington and Mesherry raise the important issue of informed consent for the treatment of people with psychosocial disabilities in Europe [34]. In other studies conducted by Sabzevari et al. among medical staff of the hospitals affiliated with the Mashhad University of Medical Sciences, the highest level of respect for patients’ rights was found in the area of respect for patients’ privacy and confidentiality, which was assessed as excellent by all respondents (100%). The lowest value of compliance with patients’ rights was associated with the presentation of adequate and appropriate information addressed to patients, which was rated excellent by 48.1% respondents [35].

Human dignity and subjectivity require that medical personnel observe the highest standards of ethical conduct and respect the intimacy of the patient. In the authors’ own study, Spanish nursing students ranked the patient’s right to respect for dignity with the highest mean values, demonstrating the importance of this right in personalized medical care.

Of all the staff caring for patients, it is nurses who spend the most time with the patients, see their behavior and recognize their needs. Thus, the quality of nursing care depends on the knowledge and experience of nurses. As Sheikhtaheri et al. proved in their study on a group of Iranian nurses, the mean score of nurses’ knowledge of patients’ rights was acceptable, while more experienced and educated nurses showed more knowledge about patients’ rights. However, compliance with patients’ rights by the nurses involved in the study was questionable [36]. In the current study, the level of nursing students’ knowledge of patients’ rights significantly varied. The highest average values were obtained by students from Spain, while the most critical in self-evaluation were those from Slovakia.

An attempt to determine the extent to which doctors and nurses in Oman were aware of the importance of patients’ rights and their observance was undertaken by Al-Saadi et al. Their research showed that overall awareness of the importance of patients’ rights among medical staff was high (91.5%), although compliance with these rights in practice was much lower (63.8%) [37].

Nurses often take on the role of the patient’s spokesperson, yet daily nursing practice also includes certain shortcomings with regard to respecting patients’ rights [38]. During their studies, nursing students acquire knowledge, clinical skills and social competences in order to fulfill their duties towards patients and their families with due diligence in their professional work.

Aydin Er et al. presented the results of a study involving 238 nursing students from the West Black Sea Universities in Turkey in which the majority of the nursing students held desirable attitudes toward patient information, truth-telling and protection of patients’ privacy and medical records. The authors proposed that ethics education, covering both patient’s rights and the obligations of nurses to defend these rights, be introduced to the study curriculum [39]. Based on a review of the literature on the topic, genuine contacts of nursing students with patients during clinical classes are of key importance in the development of the skills necessary for students working with patients. The concept of learning from patients has emerged recently, thus transferring the emphasis of learning from professionals as the example to follow to relations created between the student and the patient [40]. This development is particularly important in the domain of social competences, where nursing students should always see to the patient’s welfare, respect patient’s dignity and autonomy, display understanding for ideological and cultural differences and respond with empathy in contacts with patients and their families [20].

As indicated by Kim, in order for future nurses to be well prepared for their professional roles, it is desirable to revise the curriculum in the nursing program to strengthen interpersonal care behaviors, biomedical ethics and students’ sensitivity to human rights [41]. Moreover, mentors involved in clinical nursing education are expected to provide the optimal educational environment for achieving and demonstrating the desired level of competence in conjunction with professional ethics and patients’ rights [42]. Finally, it should be added that modern nursing entails an ethical responsibility to respect and protect patients’ rights. The presented results of the authors’ own research reflect a certain fragment of reality and provide a contribution to further scientific investigations.

## 5. Limitations and Implications for Professional Practice

The results of this study help to outline implications for professional practice. Firstly, they point to a need to analyze study programs in conjunction with assessing the effectiveness of clinical teaching in the nursing program, with particular emphasis on the courses that involve learning outcomes related to professional ethics and patients’ rights. Secondly, there is a need to disseminate the information on patients’ rights among the population in a given country. Thirdly, medical practitioners, as part of their postgraduate training, should deepen their knowledge, improve their professional skills and develop their social skills throughout their careers. This will ensure a sufficiently high level of medical care for the patient, which will translate into therapeutic safety and patient satisfaction with the medical services provided. The presented study is the first one of this type conducted on the international scale in selected European countries, i.e., Poland, Spain and Slovakia, but it has its limitations, such as the size of the surveyed group, and needs to be replicated with a larger number of respondents.

## 6. Conclusions

The degree to which patients’ rights are respected in healthcare facilities in Poland, Spain and Slovakia in the subjective assessment of nursing students is significantly different and is not always favorable for the patient.A variation in the level of nursing students’ knowledge of selected patients’ rights in personalized healthcare was observed, requiring in-depth educational activities at the university level in respective countries.The degree of knowledge of patients’ rights among nursing students is not uniform and includes the right to information on the patient’s health, the confidentiality of patient-related information and medical records, to withdraw their objection to organ and tissue donation, to pastoral care and to deposit valuable items.The right to respect dignity, which also includes the right to die in peace and dignity in personalized medical care, was rated the highest by Spanish first-year students.

## Figures and Tables

**Figure 1 jpm-11-00191-f001:**
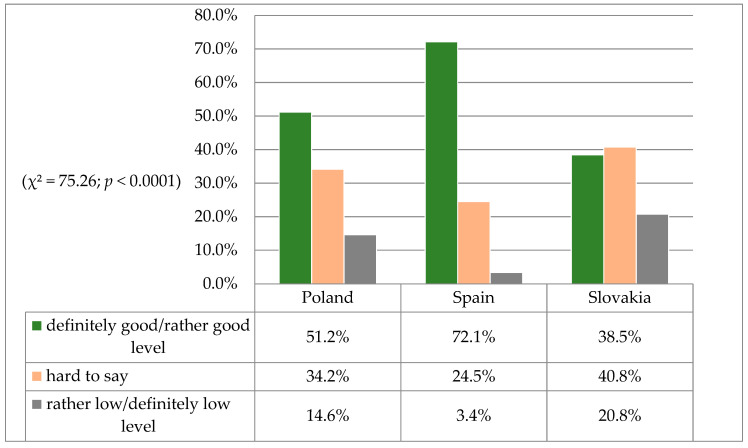
Observance of patients’ rights as perceived by nursing students.

**Figure 2 jpm-11-00191-f002:**
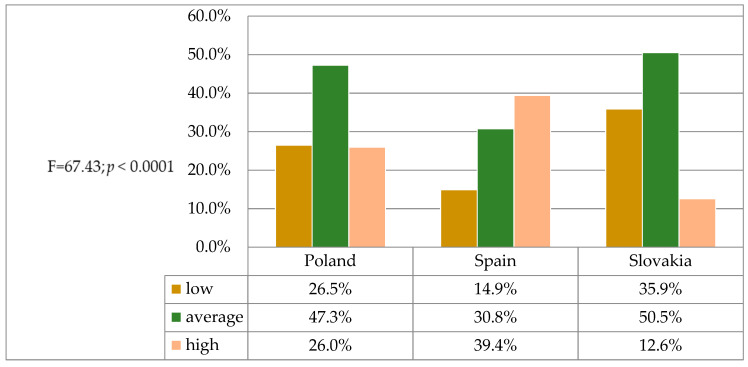
Self-assessment of the level of the students’ knowledge of patients’ rights—distribution of answers.

**Table 1 jpm-11-00191-t001:** Distribution of answers to questions on patients’ rights-a comparative analysis.

No.	Patients’ Rights	Answer Scale	Responses-Number (%)			Chi-Squared Test (χ^2^)	*p*
			Poland *n* = 404	Spain *n* = 208	Slovakia *n* = 390		
1	The right to obtain comprehensive and understandable information on their health condition	Yes	115 (28.5)	193 (92.8)	199 (51.0)	315.61	0.0001 ***
		No	92 (22.8)	9 (4.3)	109 (28.0)		
		I have no opinion	133 (32.9)	6 (2.9)	32 (8.2)		
		I don’t know	64 (15.8)	0 (0.0)	50 (12.8)		
2	The patient’s right to pastoral care while staying in hospital	Yes	262 (64.9)	118 (56.7)	210 (58.9)	122.24	0.0001 ***
		No	13 (3.2)	22 (10.6)	62 (9.7)		
		I have no opinion	62 (15.4)	67 (32.2)	50 (17.9)		
		I don’t know	67 (16.6)	1 (0.5)	68 (13.6)		
3	The right to deposit valuables in a hospital depository during on-site (stationary) treatment	Yes	310 (76.7)	130 (62.5)	356 (91.3)	121.64	0.0001 ***
		No	26 (6.4)	19 (9.1)	13 (3.3)		
		I have no opinion	43 (10.6)	59 (28.4)	10 (2.6)		
		I don’t know	25 (6.2)	0 (0.0)	11 (2.8)		
4	Data protection and confidentiality of patient information by healthcare professionals	Yes	346 (85.6)	165 (79.3)	356 (91.3)	66.13	0.0001 ***
		No	14 (3.5)	37 (17.8)	11 (2.8)		
		I have no opinion	29 (7.2)	4 (1.9)	9 (2.3)		
		I don’t know	15 (3.7)	2 (1.0)	14 (3.6)		
5	Disclosure of information subject to professional secrecy by healthcare professionals	I don’t know	59 (14,6)	20 (9.6)	89 (22.8)	36.78	0.0001 ***
		Yes, if the information covered could contribute to a risk to the health and life of others	225 (55.7)	141 (67.8)	180 (46.2)		
		Never	40 (9.9)	10 (4.8)	26 (6.7)		
		At the request of the court	80 (19.8)	37 (17.8)	95 (24.4)		
6	Obligation to provide the patient with a copy of the records of hospital/ambulatory treatment by the medical facility	Yes	289 (71.5)	156 (75.0)	264 (67.7)	51.14	0.0001 *****
		No	30 (7.4)	18 (8.7)	53 (13.6)		
		I have no opinion	42 (10.4)	34 (16.4)	29 (7.4)		
		I don’t know	43 (10.6)	0 (0.0)	44 (11.3)		
7	Discharge of a patient from a hospital upon the patient’s own request	Yes	321 (79.5)	101 (48.6)	296 (75.9)	86.00	0.0001 *****
		No	10 (2.5)	3 (1.4)	20 (5.1)		
		Only if their life is not in danger	47 (11.6)	73 (35.1)	56 (14.4)		
		I don’t know	26 (6.4)	31 (14.9)	18 (4.6)		
8	Withdrawal of the patient’s objection to organ and tissue donation	Yes	243 (60.2)	150 (72.1)	221 (56.7)	15.17	0.004 ****
		No	52 (12.9)	22 (10.6)	54 (13.9)		
		I don’t know	109 (27.0)	36 (17.3)	115 (29.5)		

Explanations: * *p* < 0.05; ** *p* < 0.01; *** *p* < 0.001.

**Table 2 jpm-11-00191-t002:** Variation in students’ self-assessed knowledge of patients’ rights.

Variables	Country of Origin			ANOVA (F)	*p* Value
	Poland *n* = 404 (40.3%)	Spain *n* = 208 (20.8%)	Slovakia *n* = 390 (38.9%)		
	M ± SD, Me, Min.–Max., 95% CI	M ± SD, Me, Min.–Max., 95% CI	M ± SD, Me, Min.–Max., 95% CI		
Self-assessment of students’ knowledge of patients’ rights (rating scale 2–5)	3.00 ± 0.73, 3.00, 2.00–5.00, 2.93 ± 3.07	3.54 ± 0.92, 4.00, 2.00–5.00, 3.42 ± 3.67	2.79 ± 0.69, 3.00, 2.00–5.00, 2.72 ± 2.86	F = 67.43	0.0001 ***

Explanations: * *p <* 0.05; ** *p <* 0.01; *** *p* < 0.001. *n*, subgroup size; M, arithmetic mean; SD, standard deviation; Me, median; Min., minimum; Max, maximum; 95% CI, confidence interval.

**Table 3 jpm-11-00191-t003:** Patients’ right to dignity—comparison of rankings.

Variables	Country of Origin			ANOVA (F)	*p* Value
	Poland *n* = 404 (40.3%)	Spain *n* = 208 (20.8%)	Slovakia *n* = 390 (38.9%)		
	M ± SD, Me, Min.–Max., 95% CI	M ± SD, Me, Min.–Max., 95% CI	M ± SD, Me, Min.–Max., 95% CI		
Ranking on a 2 to 5 scale given to the patient’s right to dignity by the respondents	3.43 ± 0.77, 4.00, 2.00–4.00, 3.35–3.50	4.37 ± 0.92, 5.00, 2.00–5.00, 4.24–4.49	3.27 ± 0.79, 3.00, 2.00–4.00, 3.19–3.35	F = 133.56	0.0001 ***

Explanations: * *p* < 0.05; ** *p* < 0.01; *** *p* < 0.001. *n*, subgroup size; M, arithmetic mean; SD, standard deviation; Me, median; Min., minimum; Max., maximum; 95% CI—confidence interval.

## Data Availability

The data presented in this study are available on request from the corresponding author.

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
