# Peer review of "Personalized Healthcare: The Importance of Patients’ Rights in Clinical Practice from the Perspective of Nursing Students in Poland, Spain and Slovakia—A Cross-Sectional Study"

_jpm, 2021, doi:10.3390/jpm11030191_

Round 1

Reviewer 1 Report

The authors aimed to define the role and importance of patients’ rights in personalized healthcare from the perspective of nursing students in Poland, Spain and Slovakia and they found that the level of students’ knowledge of patients’ rights and the respect for them by medical personnel is, in the opinion of the respondents, quite diverse and requires in-depth educational activities among students of nursing at the university level in respective countries.

Major concerns

It seems that personalized medicine has been put in the title to justify the adaptation to the scientific journal

It is a study based on surveys that offers information of little interest to the scientific community and to the advancement of science

The scientific journal to which the document is sent so that it has another profile of more scientific documents and backed by a data record and not qualitative studies

The authors have forgotten to delete their own comments in the margins and, for example, comments have been left on lines 143 and 194. This indicates that the authors have not been careful when writing the document.

Another example is page 9 in which it is literally a blank page and they have not taken into account the layout of the document

The conclusions are weak for the size of the scientific journal. It may have a better place in a less impact scientific journal

Therefore, in view of the methodology used and the conclusions obtained, I do not recommend the publication of this work in a scientific journal of these characteristics.

Author Response

Dear Reviewers, Thank you very much for a thorough editorial assessment of our manuscript, positive opinions, as well as reviewers’ remarks. We used them as an important guide to improving the quality of our paper. Our implemented corrections were done strictly according to their comments. All changes made in the text are marked in yellow and blue. We are enclosing with the re-edited manuscript and cover letter as responses to Reviewers, detailing how we followed their suggestions. Thank you very much for your kind consideration of our paper. Yours sincerely Ewa Kupcewicz, PhD The authors appreciate the opinions of the Reviewer 1 and would like to address their comments: Taking care of patients should not be limited only to the formal treatment of their disease, but also encompass a respectful approach to the personal and emotional spheres of patients. This study indicates, that this approach is different in investigated countries, and in all the three countries there is room for improvement. The findings are based on large scale data and on validated methods; they were consulted with professional statisticians to support their scientific value. This way, we believe that the results of a present study may help to make health-professionals teaching programs better. The authors used MS.WORD processing software to write the manuscript and to exchange our comments on the text. The comments may be visible, or may be switched off using menu Review Tab under show „final version” command, for reader’s convenience. The Editors moved the Table 1 to the end of the manuscript leaving a blank page. The empty space will be corrected at a final version of the paper. As suggested by the Reviewer, final linguistic corrections were carried out by the language translation office,,Lider Konsorcjum – Firma OSCAR - Foreign Language School and Translation Office Joanna Jensen with headquarters at ul. Reja 2/4 lok 1, 10-565 Olsztyn. "

Reviewer 2 Report

The topic is interesting because it knows some dimensions of future nurses. From the evidence given, the path can be best constructed. In the discussion I would add references to training and other issues (i.e. fear of treating patients, difficulties in the workplace)

Author Response

Dear Reviewers, Thank you very much for a thorough editorial assessment of our manuscript, positive opinions, as well as reviewers’ remarks. We used them as an important guide to improving the quality of our paper. Our implemented corrections were done strictly according to their comments. All changes made in the text are marked in yellow and blue. We are enclosing with the re-edited manuscript and cover letter as responses to Reviewers, detailing how we followed their suggestions. Thank you very much for your kind consideration of our paper. Yours sincerely Ewa Kupcewicz, PhD The authors appreciate the opinions of Reviewer 2 and would like to address the comments: The authors have made every effort to supplement the Discussion with more valuable information. This section has been modified.

Reviewer 3 Report

The study is well written and interesting. It focuses on patients’ rights in health care and might contribute for readers of the journal. However, there are some issues that must be corrected-improved.

The methods section is not clear at all. Particularly is not clear how the instrument was designed and validated. What type of statistical analysis was conducted to assess the reliability and validity? how was the sample defined?

Readers and practitioners require more details of the instrument design to comprehend how this study was developed.

Authors stated that “For all the tests, a significance level of p < 0.05 was assumed”, later in table 1 authors employed **p < 0.01; ***p < 0.001, this is contradictory. Please be consistent.

What is the rationale for using a Cronbach’s alpha from 0.60 to 0.71?

Author got positive opinion (No. 4/2020) of the Senate Committee on Ethics of 194 Scientific Research at the Higher School in Olsztyn, Poland. What about the study in Spain and Slovakia?

The research is focused on the degree of observing patients’ rights in health care institutions in the opinion of nursing students. However, the discussion section is scarce in similar opinions. Instead, it is mainly focused on patients’ rights from different perspectives (doctors and patients). Authors might enrich the discussion in this regard.

Authors use “healthcare” and “health care” indistinctly. Please be consistent.

More observations include:

Row 129, change the “and” by a comma
Row 171, please review commas in: knowledge, “, 4"

Row 361 Please review redaction “p < 0.59). said it could be”

Row 420, review the font style.

Author Response

Dear Reviewers, Thank you very much for a thorough editorial assessment of our manuscript, positive opinions, as well as reviewers’ remarks. We used them as an important guide to improving the quality of our paper. Our implemented corrections were done strictly according to their comments. All changes made in the text are marked in yellow and blue. We are enclosing with the re-edited manuscript and cover letter as responses to Reviewers, detailing how we followed their suggestions. Thank you very much for your kind consideration of our paper. Yours sincerely Ewa Kupcewicz, PhD The authors appreciate the opinions of Reviewer 3 and would like to address the comments: The study was carried out on specially selected groups. The authors of the paper consulted experienced statisticians (employees of the Faculty of Mathematics and Computer Science, University of Warmia and Mazury in Olsztyn) displaying extensive command of biostatistics. The selection of statistical methods and the accuracy of the conducted analyses were verified. To assess the reliability of the questionnaire, use was made of Cronbach’s alpha coefficient. Based on a review of the literature on the topic, an accurate and commonly adopted value of the Cronbach’s alpha coefficient is at least 0.6. The data in the text, tables and figures were verified. Small corrections have been made to the manuscript text. A detailed description of the tool was included in the section Research Instruments. Moreover, the set of questions results from indices included in Table 1. The process of the construction of the applied tool involved the development of a set of statements concerning the studied variables using information retrieved from the literature on the subject and available methods. Once the final set of questions in the Polish language was established, it was translated into the Spanish and Slovak languages. The research tool in equivalent language versions was subjected to further adaptation by a team of experienced statisticians (as described above). The data in the manuscript was verified and the text was slightly modified. The consistency of the questionnaire was assessed using Cronbach’s alpha coefficient. Following an in-depth statistical analysis, the value of a coefficient ranged from 0.60 to 0.71. The obtained result indicates that the measurement tool provides accurate values. Based on the literature review, the accurate and commonly used Cronbach’s alpha coefficient value is at least 0.6 [Stodolak, A., Methodology of scientific research in nursing. Higher Medical School in Legnica. Ed. I, Legnica 2011]. The authors of the manuscript have also consulted an experienced statistician team, who re-verified the selection of statistical tests and the accuracy of the conducted analyses. Empirical data used in the manuscript forms a part of a larger scientific project conducted within the framework of a foreign research internship. The Senate Committee for Research received a project including all the scientific research centers participating in the study (from Poland, Spain and Slovakia). The partners from Spain and Slovakia approved such a mode of conduct. The authors have made every effort to supplement the Discussion with more valuable information. This section has been modified. As suggested by the Reviewer, final linguistic corrections were carried out by the OSCAR Translation Office from Olsztyn (address: ul. Reja 2/4 lok 1, 10-565 Olsztyn, Poland) The data in the manuscript has been verified and small corrections have been made in the text as instructed by the Reviewer.

Round 2

Reviewer 1 Report

The changes made are insufficient and therefore my recommendation is rejection